# Classification Performance of the Ages and Stages Questionnaire: Influence of Maternal Education Level

**DOI:** 10.3390/children10030449

**Published:** 2023-02-25

**Authors:** Marguerite Lockhart, Robin Chaux, Mathilde Chevin, Magali Celle, Tiphaine Raia-Barjat, Hugues Patural, Stéphane Chabrier, Antoine Giraud

**Affiliations:** 1Neonatal Intensive Care Unit, Department of Paediatrics, Centre Hospitalier Universitaire de Saint-Etienne, 42270 Saint-Etienne, France; 2INSERM, U1059 Sainbiose, Université Jean Monnet, 42100 Saint-Etienne, France; 3Department of Clinical Research and Pharmacology, Centre Hospitalier Universitaire de Saint-Etienne, 42270 Saint-Etienne, France; 4Department of Paediatrics, McGill University, Montreal, QC H3A 0G4, Canada; 5Coordination du Réseau SEVE, Centre Hospitalier Universitaire de Saint-Etienne, 42270 Saint-Etienne, France

**Keywords:** prematurity, neurodevelopment, assessment, maternal instruction

## Abstract

(1) Background: The Ages and Stages Questionnaire—Third Edition (ASQ-3) is a parental screening questionnaire increasingly being used to evaluate the development of preterm children. We aimed to assess the classification performance of the ASQ-3 in preterm infant follow-up. (2) Methods: In this cross-sectional study, we included 185 children from the SEVE longitudinal cohort born <33 weeks of gestational age between November 2011 and January 2018, who had both an ASQ-3 score at 24 months of corrected age (CA) and a revised Brunet–Lézine (RBL) scale score at 30 months of CA. The ASQ-3 overall score and sub-scores were compared to the RBL developmental quotient (DQ) scores domain by domain. The diagnostic performance of the ASQ-3 was evaluated with the RBL as the reference method by calculating sensitivity, specificity, and positive and negative likelihood ratios. A multivariate analysis assessed the association between low maternal education level and incorrect evaluation with the ASQ-3. (3) Results: The ASQ-3 overall score had a specificity of 91%, a sensitivity of 34%, a positive likelihood ratio of 3.82, and a negative likelihood ratio of 0.72. Low maternal education level was a major risk factor for incorrectly evaluating children with the ASQ-3 (odds ratio 4.16, 95% confidence interval 1.47–12.03; *p* < 0.01). (4) Conclusions: Regarding the low sensitivity and the impact of a low maternal education level on the classification performance of the ASQ-3, this parental questionnaire should not be used alone to follow the development of preterm children

## 1. Introduction

Preterm children are at high risk of neurodevelopmental impairment [1,2]. According to the EPIPAGE-2 cohort study, the rates of severe and moderate neurodevelopmental impairments at five years of age are estimated from 28% to 12%, for children born between 24 and 26 weeks of gestational age and between 32 and 34 weeks of gestational age, respectively [1]. The early evaluation of the developmental trajectory of each preterm child is essential to identify the children to whom to propose an early developmental intervention, to promote their future activity and participation [3,4]. A meta-analysis of eight studies, including 1436 children born before 37 weeks of gestational age evaluated between three and five years of age, had shown that such early developmental intervention improved intelligence quotient at preschool age (standardised mean difference 0.43 standard deviations, 95% confidence interval (CI) from 0.32 to 0.54) [4].

The two hetero-assessment scales used by professionals for assessing neurodevelopment during the first years of life and validated in France are the revised BrunetLézine (RBL) scale and the Bayley Scales of Infant and Toddler Development—Third Edition (Bayley-III) [5,6]. The RBL is the predominant validated hetero-assessment scale used to evaluate the neurodevelopment of children at preschool age in France. It calculates a global developmental quotient (DQ) by computing four DQs for sociability, gross motor function, visuospatial coordination, and language [5]. The Bayley-III evaluates the same neurodevelopmental domains, has the same age groups, and uses a similar scoring technique as the RBL [6].

The Ages and Stages Questionnaire—Third Edition (ASQ-3) is a validated parental screening questionnaire increasingly being used to evaluate the development of preterm children [7,8]. It evaluates the same neurodevelopmental domains as RBL, in addition to the problem-solving domain, and consists of six questions for each domain of child development [8]. A single study compared the ASQ-3 to the RBL scale in this population and found a good correlation between the overall ASQ-3 score and the global DQ of the RBL in children with neurodevelopmental impairment at 24 months of corrected age (CA) [9].

The aim of our study was to assess the classification performance of the ASQ-3 in preterm children and the impact of maternal education level on the classification performances of the ASQ-3, using the RBL scale as a reference. As the ASQ-3 is a parental questionnaire, we hypothesised that the ASQ-3 has low sensitivity and that low maternal education level was associated with incorrect classification with the ASQ-3.

## 2. Materials and Methods

This cross-sectional study was performed and reported according to the STROBE guidelines for observational studies [10].

### 2.1. Population Sample

The *Suivi des Enfants Vulnérables sur le Réseau Elena* (SEVE) Network is a regional cohort ensuring a free-of-charge and standardised follow-up of vulnerable children, as described [11]. As part of the follow-up, parents were encouraged to complete the ASQ-3 at 24 months of corrected age (CA) and a systematic encounter with an experienced neuropsychologist was offered at 30 months of CA to perform an RBL assessment. All children born before 33 weeks of gestational age (GA) between November 2011 and January 2018 with an ASQ-3 evaluation and an RBL evaluation were included in our study. There were no exclusion criteria.

Data were longitudinally collected by using standardised questionnaires throughout the predefined follow-up protocol and included maternal age, maternal education level, antenatal steroids administration, multiple pregnancy, vaginal delivery versus caesarean-section, GA at birth, birth measurements, Apgar score at 1 min and 5 min, the use of surfactant, cerebral haemorrhage, periventricular leukomalacia, necrotising enterocolitis, ventilatory support duration, hospitalisation duration, and feeding at discharge. Information on the maternal education level was recorded during the neuropsychologist consultation. A low maternal education level was defined as the highest attained education below secondary education (i.e., baccalaureate degree or less in the French school system). A high education level was defined by any high school degree. Birth measurement z-scores were calculated by GA and sex using the INTERGROWTH- 21st International Newborn Size at Birth Standards application v1.3.5 (University of Oxford, UK).

### 2.2. Neurodevelopmental Assessment

Neurodevelopmental evaluation at 30 months of CA was performed by an experienced neuropsychologist blinded to the ASQ-3 results, using the RBL scale. The RBL is the predominant scale utilised to evaluate the neurodevelopment of children from 2 to 30 months of CA in France [5,12]. It was validated on a sample of 1032 children born at term [5]. A global DQ was established by computing four DQs for sociability, gross motor function, visuospatial coordination, and language. Each DQ is normalised to the CA and has a mean of 100 and a standard deviation (SD) of 15 [5]. A DQ < 85 is defined as abnormal [5].

Parents completed the French ASQ-3 adapted for children at 24 months of CA. The ASQ-3 consists of six questions for five domains of child development: communication, gross motor, fine motor, problem-solving, and personal–social [8]. A score <−2 SDs is defined as abnormal [8]. The overall ASQ-3 score is considered abnormal if at least one domain is abnormal [8].

### 2.3. Ethics

Written informed parental consent was mandatory to be included in the SEVE Network. The French National Technologies and Civil Liberties Commission (CNIL) approved data collection (authorisation no. 1530737). This study was authorised by the Ethical Committee of the Saint-Étienne University Hospital (April 2020, IRBN392020/CHUSTE).

### 2.4. Statistical Analysis

Categorical variables are expressed as absolute numbers (percentage). Continuous variables are expressed as mean (SD) or median (interquartile range) according to their distribution.

The ASQ-3 was compared to the RBL domain by domain: communication, gross motor, fine motor, personal–social domains, and overall ASQ scores were compared to language, gross motor function, visuospatial coordination, sociability, and global DQs, respectively. The classification performance of the ASQ-3 was evaluated by generating contingency tables for each score. The RBL was defined as the reference method. DQs < 85 were considered as positive, i.e., abnormal development [5]. ASQ-3 domain scores and overall ASQ score with one or more domains <−2 SDs were considered as positive, i.e., abnormal development [8]. The children considered correctly classified with the ASQ-3 were true positives and true negatives. The children considered incorrectly classified were false positives and false negatives.

Using that classification contingency table, sensitivity (true positive rate) and specificity (true negative rate) with their 95% CIs were computed. To summarize the information contained in both sensitivity and specificity, positive and negative likelihood ratios were also computed for each score using Formulas (1) and (2) [13].
(1)$$\mathrm{Positive}\;\mathrm{likelihood}\;\mathrm{ratio}=\frac{\mathrm{sensitivity}}{1-\mathrm{specificity}}=\frac{\mathrm{Pr}(T+|D+)}{\mathrm{Pr}(T+|D-)}$$
(2)$$\mathrm{Negative}\;\mathrm{likelihood}\;\mathrm{ratio}=\frac{1-\mathrm{sensitivity}}{\mathrm{specificity}}=\frac{\mathrm{Pr}(T-|D+)}{\mathrm{Pr}(T-|D-)}$$

“T+” and “T−” represent the positivity or negativity of the test, respectively. “D+” and “D− “ represent the presence or absence of abnormal development, respectively. A positive likelihood ratio > 10 would be considered satisfying, as well as a negative likelihood ratio < 0.1, whereas positive or negative likelihood ratios = 1 would mean that the test does not discriminate against individuals with or without abnormal development.

Diagnostic odds ratios with their 95% CI were also calculated to describe the odds of having a positive ASQ-3 test in children with abnormal development relative to the odds of a positive ASQ-3 test in children with normal development. The calculation of the diagnostic odds ratio and its relations to positive and negative likelihood ratios is described in Formula (3) [13].
(3)$$\mathrm{Diagnostic}\;\mathrm{odds}\;\mathrm{ratio}=\frac{\mathrm{positive}\;\mathrm{likelihood}\;\mathrm{ratio}}{\mathrm{negative}\;\mathrm{likelihood}\;\mathrm{ratio}}$$

Then, the performance of the ASQ-3 diagnostic was assessed using the area under curve (AUC) of receiver operating characteristic (ROC) curves and their 95% CIs, by means of 10,000 iterations of bootstrap. A secondary analysis stratified on maternal education level was also performed using the same methodology.

Univariate and multivariate logistic regression models were used to estimate the independent association between the classification performance of ASQ-3 and low maternal education level. The dependent variable ASQ-3 was dichotomized as previously and treated as a binary variable, with an overall ASQ-3 score < −2 SDs considered positive and an overall ASQ-3 score ≥ −2 SDs considered negative. The multivariate model was built a priori, using known potential confounders according to literature data and experts’ opinions. Sex, GA, hospitalization duration, and ventilatory support duration were included in the multivariate model as adjustment covariates as potential confounders. Quantitative covariates were coded as continuous covariates.

The results are reported as odds ratios (ORs) and their 95% CIs. All tests were two-sided, with a *p*-value <0.05 considered as statistically significant. Missing data represented 2.5% of the data and were not considered in the analysis. Analyses were performed using R software version 4.0.3 (R Foundation for Statistical Computing, Vienna, Austria).

## 3. Results

One hundred and eighty-five children had an RBL and ASQ-3 evaluation and were included in the current study. The features of the population sample are described in Table 1. The mean CA at RBL evaluation was 29.0 months with an SD of 2.7.

The results of the neurodevelopmental assessments by ASQ and RBL are summarised in Table 2. The included children had a median overall ASQ-3 score of 265, with an IQR from 235 to 285, and a mean global DQ score of 91.5, with an SD of 10.2. Twenty-six (14.3 %) children had an abnormal developmental evaluation with the ASQ-3, whereas 36 (19.7%) children had an abnormal developmental evaluation with the RBL scale. The ASQ-3 evaluation identified 8 (4.3%) children with abnormal gross motor abilities, 16 (8.6%) children with abnormal personal–social abilities, 10 (5.4%) children with abnormal communication abilities, 6 (3.2%) children with abnormal fine motor abilities, and 15 (8.2%) children with abnormal problem-solving abilities. According to the RBL evaluation, 35 (18.9%) children had an abnormal gross motor development, 38 (20.5%) children had an abnormal personal–social development, 68 (37.2%) children had an abnormal communication development, and 35 (18.9%) children had an abnormal fine motor development (Table 2).

The classification performance of the ASQ-3 is detailed in Table 3. The overall ASQ-3 score had a sensitivity of 0.34 (95% CI from 0.27 to 0.41) and a specificity of 0.91 (95% CI from 0.87 to 0.95). The cross-tabulation of the overall ASQ score and the global DQ score found a positive likelihood ratio of 3.82, a negative likelihood ratio of 0.72, a diagnostic odds ratio of 5.29 (95% CI from 4.43 to 6.17), and an area under curve of 0.67 (95% CI from 0.56 to 0.77). Regarding domain-by-domain evaluation, ASQ-3 had a sensitivity of 0.11 (95% CI from 0.07 to 0.16) for the gross motor domain, 0.08 (95% CI from 0.04 to 0.12) for the personal–social domain, 0.13 (95% CI from 0.08 to 0.18) for the communication domain, and 0.09 (95% CI from 0.05 to 0.13) for the fine motor domain. The ASQ-3 specificity was 0.97 (95% CI from 0.95 to 0.99) for the gross motor domain, 0.98 (95% CI from 0.96 to 0.99) for the personal–social domain, 0.99 (95% CI from 0.98 to 1.00) for the communication domain, and 0.98 (95% CI from 0.96 to 1.00) for the fine motor domain. The ASQ-3 positive likelihood ratios were 4.29, 3.87, 15.22, and 4.29 for the gross motor domain, the personal–social domain, the communication domain, and the fine motor domain, respectively. The ASQ-3 negative likelihood ratios were 0.72, 0.91, 0.94, 0.88, and 0.93 for the gross motor domain, the personal–social domain, the communication domain, and the fine motor domain, respectively. ASQ-3 had a diagnostic odds ratio of 4.71 (95% CI from 3.29 to 6.13) for the gross motor domain, 4.11 (95% CI from 2.49 to 5.74) for the personal–social domain, 17.39 (95% CI from 15.32 to 19.46) for the communication domain, and 4.59 (95% CI from 2.96 to 6.22) for the fine motor domain. The ASQ-3 area under curve was 0.66 (95% CI from 0.48 to 0.85) for the gross motor domain, 0.65 (95% CI from 0.43 to 0.87) for the personal–social domain, 0.78 (95% CI from 0.68 to 0.88) for the communication domain, and 0.66 (95% CI from 0.44 to 0.88) for the fine motor domain (Table 3).

The subgroup analysis of the classification performance of the ASQ-3 in children with a low maternal education level is detailed in Table 4. The overall ASQ-3 score had a sensitivity of 0.35 (95% CI from 0.21 to 0.49) and a specificity of 0.88 (95% CI from 0.78 to 0.97) in children with a low maternal education level. In these children, the cross-tabulation of the overall ASQ score and the global DQ score found a positive likelihood ratio of 2.80, a negative likelihood ratio of 0.74, a diagnostic odds ratio of 3.77 (95% CI from 2.38 to 5.16), and an area under curve of 0.66 (95% CI from 0.49 to 0.83) (Table 4).

The subgroup analysis of the classification performance of the ASQ-3 in children with a high maternal education level is detailed in Table 5. In these children, the overall ASQ-3 score had a sensitivity of 0.36 (95% CI from 0.27 to 0.44) and a specificity of 0.92 (95% CI from 0.87 to 0.97). In children with a high maternal education level, the ASQ-3 also had a positive likelihood ratio of 4.37, a negative likelihood ratio of 0.70, a diagnostic odds ratio of 6.23 (95% CI from 4.98 to 7.49), and an area under curve of 0.64 (95% CI from 0.50 to 0.77) (Table 5).

The results of the analyses calculating the association between the classification performance of ASQ-3 and a low maternal education level are displayed in Table 6. A Low maternal education level was a major risk factor for incorrectly evaluating children with the ASQ-3 on multivariate analysis (OR = 4.16, 95% CI from 1.47 to 12.03; *p* < 0.01) (Table 6).

## 4. Discussion

Compared to the RBL, the ASQ-3 had excellent specificity and low sensitivity to assess the neurodevelopment of preterm children. A low maternal education level was a major risk factor for incorrectly evaluating children with the ASQ-3.

This association between the maternal education level and the classification performance of the ASQ-3 is a significant result to consider because a low maternal education level is also a risk factor for impaired neurodevelopmental outcomes in preterm children [14,15]. Altogether, a low maternal education level can be considered as a double-risk factor in preterm children: a risk factor for an impaired neurodevelopmental outcome, and a risk factor for having such neurodevelopmental outcome incorrectly evaluated if the ASQ-3 is used alone to assess the neurodevelopment [14,15].

The Bayley-III and RBL scales evaluate the same neurodevelopmental domains, have the same age groups, and use a similar scoring technique. The accuracy of the RBL scale compared to the Bayley-III scale is 86.4% for motor skills, 90.9% for cognitive skills, 94.3% for communication skills, and 88.6% for sociability skills [16]. Our findings for the specificity and sensitivity of the ASQ-3 agree with most studies comparing the ASQ-3 to the Bayley-III at this age [17,18,19,20,21]. The ASQ-3 is a screening tool rather than a diagnostic assessment [8]. Some studies considered that the ASQ-3 is not reliable for screening neurodevelopment in children [22,23]. One prospective study considered the ASQ-3 equivalent to the Bayley-III at 2 years of CA to detect moderate to severe neurodevelopmental impairment [24]. A recent meta-analysis including 36 studies concluded that the ASQ-3 performance to identify developmental delay in children aged between 12 and 60 months was moderate, with a pooled sensitivity of 0.77 (95% CI, 0.64–0.86), a pooled specificity of 0.81 (95% CI, 0.75–0.86), a pooled positive likelihood ratio of 4.10 (95% CI, 3.17–5.30), and a pooled negative likelihood ratio of 0.28 (95% CI, 0.18–0.44) [25].

In France, the ASQ-3 has been used as an evaluating method based on a study that compared the ASQ-3 scores and RBL DQ scores at 24 months of CA [9]. Compared to our study, the previous study included more patients and found good sensitivity (88%) and specificity (57%) for the ASQ-3 [9]. The high sensitivity might be explained by the RBL being administered earlier, at 24 months of CA, because the neurodevelopmental trajectory becomes complex with age [26]. Additionally, the authors compared the overall ASQ-3 score to the global RBL DQ and did not focus on each domain of the ASQ-3 [9]. Domain-by-domain evaluation allows for more accurate identification of impairment. Besides language skills, which are reported to more strongly predict impairment than other cognitive domains, early developmental milestones before age 24 months poorly predict neurodevelopmental impairment at school age [27]. This is particularly important for fine motor skills, which involve complex neurodevelopmental functions [28].

Although the parental point of view is essential, a professional assessment of the neurodevelopmental trajectory of preterm children is critical. Parent and professional views are different; the threshold level of impairment can differ: professionals tend to prioritize functional goals, whereas parents tend to prioritize activity and participation [29]. Additionally, a professional assessment is limited to a short time window of evaluation, whereas parental evaluation reflects on an average function over time. Alongside the neurodevelopmental trajectory evaluation, a follow-up of the quality of life is also essential in this population [30,31].

Our study is the first to compare the ASQ-3 and RBL scales domain by domain and the second to compare those two neurodevelopmental assessment scales. A limitation is the population sample size of 185, but the features of this sample are comparable to those of the whole population included in the SEVE Network [11].

## 5. Conclusions

The ASQ-3 has a low sensitivity to assess the neurodevelopment of preterm children A low maternal education level is a major risk factor for incorrectly evaluating children with the ASQ-3 at 24 months of CA. The ASQ-3 should not be used alone to follow the development of preterm children. Although the parental point of view is essential, a hetero-assessment by a professional using a validated scale is critical to evaluate the neurodevelopment of preterm children.

## Figures and Tables

**Table 1 children-10-00449-t001:** Features of included children.

Features	Population (n = 185)
Maternal age, years, mean (SD)	30.5 (5.1)
Multiple pregnancy, n (%) ^a^	43 (23.4)
Antenatal steroids administration	
Incomplete, n (%)	27 (15.5)
Complete, n (%)	125 (71.8)
Vaginal delivery, n (%)	168 (70.6)
GA, weeks, mean (SD)	30.6 (2.9)
Birth weight, g, mean (SD)	1440 (550)
Birth weight z score, mean (SD)	−0.24 (0.99)
Birth height, cm, mean (SD)	39.0 (4.7)
Birth height z score, mean (SD)	−0.49 (1.25)
Birth HC, cm, mean (SD)	27.7 (3.1)
Birth HC z score, mean (SD)	−0.12 (1.21)
Male sex, n (%)	104 (56.2)
Apgar score at 1 min < 7, n (%)	82 (44.8)
Apgar score at 5 min < 7, n (%)	21 (11.7)
Surfactant administration	
1 dose, n (%)	69 (43.9)
2 doses, n (%)	13 (8.3)
Hospitalisation duration, days, mean (SD)	57.9 (27.9)
VS duration, weeks, median (IQR)	1 (0–10)
Cerebral haemorrhage, n (%)	
Grade 1, n (%)	6 (3.2)
Grade 2, n (%)	3 (1.6)
Grade 3, n (%)	4 (2.2)
Grade 4, n (%)	1 (0.5)
Necrotizing enterocolitis, n (%)	5 (2.7)
Feeding at discharge	
Breastfeeding, n (%)	49 (27.1)
Artificial, n (%)	100 (55.2)
Mixed, n (%)	32 (17.1)
Low maternal educational level, n (%)	45 (26)
High maternal educational level, n (%)	128 (74)

^a^ Indicates more than one foetus (i.e., twins, triplets). Mean, SD, median, IQR, and percentages were calculated after the exclusion of missing data. Abbreviations: GA, gestational age; HC, head circumference; IQR, interquartile range; n (%), number of patients (percentage); SD, standard deviation; VS, ventilatory support.

**Table 2 children-10-00449-t002:** Neurodevelopmental assessments.

Domains	Ages and Stages Questionnaire (ASQ)—Third Edition *	Revised Brunet–Lézine (RBL) Scale *
Overall score	265 (235–285)	91.5 (10.2)
Abnormal overall test, n (%)	26 (14.3)	36 (19.7)
Gross motor score	60 (50–60)	92.9 (11.7)
Abnormal gross motor, n (%)	8 (4.3)	35 (18.9)
Personal–social score	50 (45–60)	93.5 (12.2)
Abnormal personal–social, n (%)	16 (8.6)	38 (20.5)
Communication score	55 (50–60)	87.1 (13.1)
Abnormal communication, n (%)	10 (5.4)	68 (37.2)
Fine motor score	50 (50–60)	92.8 (11.4)
Abnormal fine motor, n (%)	6 (3.2)	35 (18.9)
Problem-solving score	50 (45–60)	NA
Abnormal problem-solving, n (%)	15 (8.2)	NA

* ASQ scores are expressed as median (IQR); RBL DQs are expressed as mean (SD). Abbreviations: DQ, developmental quotient; IQR, interquartile range; n (%), number of patients (percentage); SD, standard deviation.

**Table 3 children-10-00449-t003:** Classification performance of the Ages and Stages Questionnaire—Third Edition.

Domains	Overall	Gross Motor	Personal–Social	Communication	Fine Motor
Sensitivity, % (95% CI)	0.34 (0.27–0.41)	0.11 (0.07–0.16)	0.08 (0.04–0.12)	0.13 (0.08–0.18)	0.09 (0.05–0.13)
Specificity, % (95% CI)	0.91 (0.87–0.95)	0.97 (0.95–0.99)	0.98 (0.96–0.99)	0.99 (0.98–1.00)	0.98 (0.96–1.00)
Positive likelihood ratio	3.82	4.29	3.87	15.22	4.29
Negative likelihood ratio	0.72	0.91	0.94	0.88	0.93
DOR (95% CI)	5.29 (4.43–6.17)	4.71 (3.29–6.13)	4.11 (2.49–5.74)	17.39 (15.32–19.46)	4.59 (2.96–6.22)
AUC (95% CI)	0.67 (0.56–0.77)	0.66 (0.48–0.85)	0.65 (0.43–0.87)	0.78 (0.68–0.88)	0.66 (0.44–0.88)

Abbreviations: 95% CI, 95% confidence interval; AUC, area under curve; DOR, diagnostic odds ratio.

**Table 4 children-10-00449-t004:** Subgroup analysis of the classification performance of the Ages and Stages Questionnaire—Third Edition in children with low maternal education level.

Domains	Overall	Gross Motor	Personal–Social	Communication	Fine Motor
Sensitivity, % (95% CI)	0.35 (0.21–0.49)	0.14 (0.04–0.25)	0.20 (0.08–0.32)	0.17 (0.06–0.28)	0.14 (0.04–0.25)
Specificity, % (95% CI)	0.88 (0.78–0.97)	0.92 (0.84–0.99)	0.97 (0.91–1)	NA	0.90 (0.82–0.99)
Positive likelihood ratio	2.80	1.81	6	NA	1.48
Negative likelihood ratio	0.74	0.93	0.83	0.83	0.95
DOR (95% CI)	3.77 (2.38–5.16)	1.94 (NA–4.32)	7.25 (4.95–9.55)	NA	1.56 (NA–3.40)
AUC (95% CI)	0.66 (0.49–0.83)	0.55 (0.30–0.80)	0.73 (0.47–0.98)	0.69 (0.62–0.77)	0.55 (0.29–0.80)

Abbreviations: 95% CI, 95% confidence interval; AUC, area under curve; DOR, diagnostic odds ratio; NA, non-available.

**Table 5 children-10-00449-t005:** Subgroup analysis of the classification performance of the Ages and Stages Questionnaire—Third Edition in children with a high maternal education level.

Domains	Overall	Gross Motor	Personal–Social	Communication	Fine Motor
Sensitivity, % (95% CI)	0.36 (0.27–0.44)	0.08 (0.03–0.12)	NA	0.08 (0.04–0.13)	0.06 (0.01–0.09)
Specificity, % (95% CI)	0.92 (0.87–0.97)	0.99 (0.97–1)	0.99 (0.97–1)	0.99 (0.97–1)	NA
Positive likelihood ratio	4.37	7.85	NA	7.58	NA
Negative likelihood ratio	0.70	0.93	1.01	0.93	0.94
DOR (95% CI)	6.23 (4.98–7.49)	8.42 (5.99–10.84)	NA	8.18 (5.90–10.46)	NA
AUC (95% CI)	0.64 (0.50–0.77)	0.74 (0.41–0.99)	0.41 (NA–NA)	0.97 (0.49–0.99)	0.93 (NA–NA)

Abbreviations: 95% CI, 95% confidence interval; AUC, area under curve; DOR, diagnostic odds ratio; NA, non-available.

**Table 6 children-10-00449-t006:** Factors associated with incorrect classification with the Ages and Stages Questionnaire—Third Edition.

Variables	Univariate Analysis		Multivariate Analysis *	
	OR (95% CI)	*p*-Values	OR (95% CI)	*p*-Values
Low maternal education level	3.37 (1.52–7.47)	< 0.01	4.16 (1.47–12.03)	< 0.01
Gestational age	0.98 (0.96–1.00)	0.13	1.01 (0.97–1.05)	0.59
Male sex	0.85 (0.41–1.77)	0.65	1.06 (0.39–2.99)	0.92
Hospitalisation duration	1.01 (0.99–1.02)	0.11	1.02 (0.99–1.06)	0.15
VS duration	1.01 (0.97–1.04)	0.59	0.98 (0.93–1.02)	0.42

Abbreviations: 95% CI, 95% confidence interval; OR, odds ratio; VS, ventilatory support. * Adjusted for low maternal education level, gestational age, male sex, hospitalisation duration, and VS duration.

## Data Availability

Anonymised data are available from the corresponding authors upon reasonable request.

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
