# Peer review of "Classification Performance of the Ages and Stages Questionnaire: Influence of Maternal Education Level"

_children, 2023, doi:10.3390/children10030449_

Round 1
Reviewer 1 Report (New Reviewer)
The paper is generally well written, clear, and well structured. The authors have applied STROBE guideline to conduct this prediction model study. I few comments which the authors may address.
1) The likelihood ratios should be reported in the abstract. The likelihood ratios are probably conveying more meaning in diagnostic accuracy studies.
2) It would be helpful if the authors could provide justification for their hypothesis in page 2 (Sentence 47 and 48)
3) Sentence no 39/40: I would probably tone down the sentence a little bit: ASQ3 is being increasingly used as a ‘screening tool’ and it’s probably not a replacement for routine tools like Bayley.
4) Were the RBL assessors blinded to ASQ3 scores? This information should be available to the readers in the method section.
5) Sentence number 102 on page 3 “The overall ASQ score with one or more domains < –2 SDs was considered positive” is difficult to understand and could be better phrased.
6) Kindly quote a reference for sentence no 116-117 in page 3 (PLR and NLR)
7) How many babies underwent RBL or ASQ evaluation in total? It should be mentioned in the result section.
8) Table 1: Low maternal education level (%) is miscalculated, if total sample size is 185 it should be 24.32 until unless there are missing samples, in that case it should be mentioned.
9) Table 2: “Overall score” has been repeated.
10) It would be useful if the authors could add diagnostic odds ratio and area under curve analysis along with specificity, sensitivity, PLR and NLR.
11) The authors could conduct a subgroup analysis of low and high maternal educational level separately and report the sensitivity, specificity, PLR, NLR, AUC in either table 3 or in a separate table. This will be a very useful information for the readers.
12) There is a recently published systematic review and meta-analysis on this subject (Citation below), the authors could consider comparing their results with this review in the discussion section.
Muthusamy S, Wagh D, Tan J, Bulsara M, Rao S. Utility of the Ages and Stages Questionnaire to Identify Developmental Delay in Children Aged 12 to 60 Months: A Systematic Review and Meta-analysis. JAMA Pediatr. 2022 Oct 1;176(10):980-989. doi: 10.1001/jamapediatrics.2022.3079. Erratum in: JAMA Pediatr. 2022 Oct 17;: PMID: 36036913; PMCID: PMC9425289.
Author Response
Dear reviewers,
We thank you for having carefully read our manuscript and proposed useful comments. Please find below our answers to these comments.
Our response to reviewer #1:
1) The likelihood ratios should be reported in the abstract. The likelihood ratios are probably conveying more meaning in diagnostic accuracy studies.
Thank you. We added the likelihood ratios in the abstract as requested.
2) It would be helpful if the authors could provide justification for their hypothesis in page 2 (Sentence 47 and 48)
As requested, we provided a justification for our hypothesis (p2, line 47-48).
3) Sentence no 39/40: I would probably tone down the sentence a little bit: ASQ3 is being increasingly used as a ‘screening tool’ and it’s probably not a replacement for routine tools like Bayley.
As requested, we toned down this sentence (p1, line 39-41).
4) Were the RBL assessors blinded to ASQ3 scores? This information should be available to the readers in the method section.
Yes, the neuropsychologist performing the RBL was blinded to the ASQ-3 results. We added this information in the method section as requested (p2, line 76).
5) Sentence number 102 on page 3 “The overall ASQ score with one or more domains < –2 SDs was considered positive” is difficult to understand and could be better phrased.
Thank you for this comment. In general, a positive test is a test identifying the presence of the disease it searched for. Here, a positive test identifies an abnormal development. We modified this sentence to be better phrased, as suggested (p3, lines 103-105).
6) Kindly quote a reference for sentence no 116-117 in page 3 (PLR and NLR)
Thank you, the citation #12 Eusebi P. Diagnostic accuracy measures. Cerebrovasc Dis 2013;36: 267-272 was added as suggested.
7) How many babies underwent RBL or ASQ evaluation in total? It should be mentioned in the result section.
We do agree that this result would be of interest. Unfortunately, we do not have access to this particular data.
8) Table 1: Low maternal education level (%) is miscalculated, if total sample size is 185 it should be 24.32 until unless there are missing samples, in that case it should be mentioned.
Thank you. We added in the legend that Mean, SD, median, IQR, and percentages were calculated after the exclusion of missing data (p4, lines 157-158).
9) Table 2: “Overall score” has been repeated.
Thank you for this comment. We corrected this mistake.
10) It would be useful if the authors could add diagnostic odds ratio and area under curve analysis along with specificity, sensitivity, PLR and NLR.
Thank you. Diagnostic odds ratios and area under curve analyses were added, as requested.
11) The authors could conduct a subgroup analysis of low and high maternal educational level separately and report the sensitivity, specificity, PLR, NLR, AUC in either table 3 or in a separate table. This will be a very useful information for the readers.
Thank you. We added this subgroup analysis.
12) There is a recently published systematic review and meta-analysis on this subject (Citation below), the authors could consider comparing their results with this review in the discussion section.
Thank you for this suggestion. We added a paragraph to compare our results with those of this systematic review (p7, lines 242-246).
Reviewer 2 Report (New Reviewer)
This current paper is very interesting, it assessed the classification performance of the ASQ-3 in preterm infant follow-up. The topic is well selected, the idea is very attractive, and the design is relatively clear, but it needs further explanation. The most important thing is that the reliability and validity of the study should be further introduced and further explored. What’s more, the language needs to be further modified before I would like to finish reading the whole text.
Abstract: Line 14 to 15 should clearly introduce the research background. Line 16 should clearly introduce the research goal, the design and the theoretical questions relating to the target research goal. What’s more proofreading should made on the use of language.
Introduction: The paper should explain the background of the study in detail, for example, the use of different scales in accessing neurodevelopment, and why the current study is carried out. At the end of the introduction, the article should clearly introduce the research goal and research hypothesis in relation to the central target research question.
Method: The article should clearly report the participants’ information, for example, number, sex, age, education and so on. The description of the task design should be added. The assessment tool should be introduced including reliability of Questionnaires. Questionnaire invented or modified in the article or as supplementary materials should be attached by the end of the paper.
Research result: The results should be explained in detail table by table.
Discussion: The summary of the recent findings should be clearly introduced at the beginning of the discussion section. Further discussion on the analysis should be introduced with analytical results, for example, paragraph 2 of the 4.Discussion.This current paper is very interesting, it assessed the classification performance of the ASQ-3 in preterm infant follow-up. The topic is well selected, the idea is very attractive, and the design is relatively clear, but it needs further explanation. The most important thing is that the reliability and validity of the study should be further introduced and further explored. What’s more, the language needs to be further modified before I would like to finish reading the whole text.
Abstract: Line 14 to 15 should clearly introduce the research background. Line 16 should clearly introduce the research goal, the design and the theoretical questions relating to the target research goal. What’s more proofreading should made on the use of language.
Introduction: The paper should explain the background of the study in detail, for example, the use of different scales in accessing neurodevelopment, and why the current study is carried out. At the end of the introduction, the article should clearly introduce the research goal and research hypothesis in relation to the central target research question.
Method: The article should clearly report the participants’ information, for example, number, sex, age, education and so on. The description of the task design should be added. The assessment tool should be introduced including reliability of Questionnaires. Questionnaire invented or modified in the article or as supplementary materials should be attached by the end of the paper.
Research result: The results should be explained in detail table by table.
Discussion: The summary of the recent findings should be clearly introduced at the beginning of the discussion section. Further discussion on the analysis should be introduced with analytical results, for example, paragraph 2 of the 4.Discussion.This current paper is very interesting, it assessed the classification performance of the ASQ-3 in preterm infant follow-up. The topic is well selected, the idea is very attractive, and the design is relatively clear, but it needs further explanation. The most important thing is that the reliability and validity of the study should be further introduced and further explored. What’s more, the language needs to be further modified before I would like to finish reading the whole text.
Abstract: Line 14 to 15 should clearly introduce the research background. Line 16 should clearly introduce the research goal, the design and the theoretical questions relating to the target research goal. What’s more proofreading should made on the use of language.
Introduction: The paper should explain the background of the study in detail, for example, the use of different scales in accessing neurodevelopment, and why the current study is carried out. At the end of the introduction, the article should clearly introduce the research goal and research hypothesis in relation to the central target research question.
Method: The article should clearly report the participants’ information, for example, number, sex, age, education and so on. The description of the task design should be added. The assessment tool should be introduced including reliability of Questionnaires. Questionnaire invented or modified in the article or as supplementary materials should be attached by the end of the paper.
Research result: The results should be explained in detail table by table.
Discussion: The summary of the recent findings should be clearly introduced at the beginning of the discussion section. Further discussion on the analysis should be introduced with analytical results, for example, paragraph 2 of the 4.Discussion.
Author Response
Dear reviewers,
We thank you for having carefully read our manuscript and proposed useful comments. Please find below our answers to these comments.
Our response to reviewer #2:
1) Abstract: Line 14 to 15 should clearly introduce the research background. Line 16 should clearly introduce the research goal, the design and the theoretical questions relating to the target research goal. What’s more proofreading should made on the use of language.
Thank you for this comment. The research background is introduced in the Background section of the Abstract: “The Ages and Stages Questionnaire – Third Edition (ASQ-3) is a parental screening questionnaire increasingly being used to evaluate the development of preterm children” (p1, lines 14-15). The research goal is written before the Methods section of the Abstract: “We aimed to assess the classification performance of the ASQ-3 in preterm infant follow-up” (p1, line 16). An editing of the English language was performed by a native speaker.
2) Introduction: The paper should explain the background of the study in detail, for example, the use of different scales in accessing neurodevelopment, and why the current study is carried out. At the end of the introduction, the article should clearly introduce the research goal and research hypothesis in relation to the central target research question.
Thank you for this comment. The use of different scales in accessing neurodevelopment is explained in the first paragraph of the Introduction section (p1, lines 33-44). The research goal and hypothesis are stated in the last paragraph of the Introduction section: “The aim of our study was to assess the classification performance of the ASQ-3 in preterm children and the impact of maternal education level on the classification performances of the ASQ-3, using the RBL scale as a reference. Since the ASQ-3 is a parental questionnaire, we hypothesised that the ASQ-3 has low sensitivity and that low maternal education level was associated with incorrect classification with the ASQ-3” (p2, lines 45-49).
3) Method: The article should clearly report the participants’ information, for example, number, sex, age, education and so on. The description of the task design should be added. The assessment tool should be introduced including reliability of Questionnaires. Questionnaire invented or modified in the article or as supplementary materials should be attached by the end of the paper.
Thank you for this comment. These participants’ features are detailed in Table 1. We do not have the ASQ questionnaire to our article due to copyright laws. Nevertheless, it can be accessed through the reference #7 Squires J, Bricker DD, Twombly E, et al. Ages & Stages Questionnaires: A Parent-Completed, Child-Monitoring System (3rd ed.). Paul H. Brookes Publishing, 2009.
4) Research result: The results should be explained in detail table by table.
Thank you for this suggestion. As requested, we explained our results in detail, table by table, throughout the Results section.
5) Discussion: The summary of the recent findings should be clearly introduced at the beginning of the discussion section. Further discussion on the analysis should be introduced with analytical results, for example, paragraph 2 of the 4.
Thank you for this comment. The summary of the recent findings is clearly introduced at the beginning of the discussion section: “Compared to the RBL, the ASQ-3 had excellent specificity and low sensitivity to assess neurodevelopment of preterm children. A low maternal education level was a major risk factor of incorrectly evaluating children with the ASQ-3” (p7, lines 225-227).
Round 2
Reviewer 2 Report (New Reviewer)
This study examined the effectiveness of ASQ-3, and pointed out the weakness of low sensitivity in assessing neurodevelopment of preterm children, which is significant for the diagnosis and also sets out requests for the future improvement of the instrument.
Except minor language problem, for example, in line270, there should be a period before A, which needs to be further revised, the article has reached the standard of publishing.
This manuscript is a resubmission of an earlier submission. The following is a list of the peer review reports and author responses from that submission.
Round 1
Reviewer 1 Report
I appreciate the opportunity to review the article entitled “Low maternal education level is associated with incorrect evaluation with the Ages and Stages Questionnaire” presented to Children for possible publication. This study aims “to assess the classification performance of the ASQ-3 in preterm children and the impact of maternal education level, using the RBL scale as a reference”. The preterm children were assessed at 24 months with ASQ-3 and at 30 months with RBL. The authors observed good specificity, but low sensitivity between the two instruments, and pointed out the low maternal education level as a greater risk factor for incorrect assessment in the ASQ-3.
Major revisions are needed. Here are some suggestions:
Overall:
The authors stated the low maternal education level “has a high impact on the neurodevelopment of preterm children” and “this finding is a major result to consider”, but is unclear if this association is with the child performance or the correct answer in the ASQ-3. The entire manuscript is unclear about that. Also, there is no discussion about low maternal education level and the found association.
Title:
The title does not give the idea that an analysis was performed to verify the sensitivity and specificity of the instrument.
Aims:
The aims are not aligned to the analysis, results and discussion.
Methods:
-
I suggest further describing which analyzes were performed for sensitivity and specificity.
-
I suggest further describing the dependent variable used to logistic regression models. Was the RBL performance? Was the answer correct or incorrect on ASQ-3?
Results:
-
I suggest further describing the dependent variable used to logistic regression models. Was the RBL performance? Was the answer correct or incorrect on ASQ-3? How many were corrected and how many incorrected?
Discussion:
The discussion is based on the sensitivity and specificity of the instruments, there is nothing about maternal education level.
Author Response
The authors stated the low maternal education level “has a high impact on the neurodevelopment of preterm children” and “this finding is a major result to consider”, but is unclear if this association is with the child performance or the correct answer in the ASQ-3. The entire manuscript is unclear about that. Also, there is no discussion about low maternal education level and the found association.
Thank you very much for these remarks allowing us to improve our manuscript. Our study is about the performance of the ASQ-3. To improve clarity, we changed the vocable “performance of the ASQ-3” to “classification performance of the ASQ-3” throughout the manuscript. We also enhanced the discussion regarding maternal education level accordingly.
The title does not give the idea that an analysis was performed to verify the sensitivity and specificity of the instrument.
Thank you for this suggestion. We changed the title to “Classification performance of the Ages and Stages Questionnaire: influence of maternal education level” to better describe the study.
The aims are not aligned to the analysis, results and discussion.
Thank you for this comment. We revised the Methods, Results, and Discussion section to better describe our study.
I suggest further describing which analyzes were performed for sensitivity and specificity.
Thank you. As requested, these analyses were further described in the method section.
I suggest further describing the dependent variable used to logistic regression models. Was the RBL performance? Was the answer correct or incorrect on ASQ-3?
Thank you. As requested, the performed analyses were further described in the method section.
I suggest further describing the dependent variable used to logistic regression models. Was the RBL performance? Was the answer correct or incorrect on ASQ-3? How many were corrected and how many incorrected?
Thank you. We further described our results, as requested.
The discussion is based on the sensitivity and specificity of the instruments, there is nothing about maternal education level.
Thank you for this comment. We added a new paragraph in the discussion section to better discuss our results regarding maternal education level.
Reviewer 2 Report
The first paragraph under discussion is very critical since it summarizes the study findings. The way it is currently written is very confusing. It would be helpful if the message is limited to what you conclude based on the results. Right now, I am confused which information is from literature and which information is based on your study within this paragraph.
Author Response
Thank you for this useful comment. We enhanced and split this critical paragraph, to better synthetise our results and discuss them with the current literature.
Reviewer 3 Report
Revise title and conclusion
Author Response
Thank you for this suggestion. The title was changed to “Classification performance of the Ages and Stages Questionnaire: influence of maternal education level”, and the conclusion was revised accordingly.
Round 2
Reviewer 1 Report
The manuscript still presents the idea of associating maternal education level with child development, especially in the title and objective. It is unclear whether the idea associates maternal education level with the mother's incorrect answers about child development or the analysis with other instruments.
Also, the idea presented as the main result in the discussion and conclusion is about maternal education level associated with incorrect answers about child development. However, only one paragraph of discussion is about this, all the rest of the discussion is about the comparison between the two instruments used.
Further, considering low maternal education level is associated with incorrect answers in ASQ-3, the conclusion should state this instrument could not be adequate for THIS population, and not a general population. What are the results for subjects with a high education level?
Author Response
The manuscript still presents the idea of associating maternal education level with child development, especially in the title and objective. It is unclear whether the idea associates maternal education level with the mother's incorrect answers about child development or the analysis with other instruments.
Dear reviewer, we are sorry that this revised version didn’t match your expectations. We changed the vocable “performance of the ASQ-3” to “classification performance of the ASQ-3” throughout the manuscript, and we clarified that we assessed the performance of the ASQ-3 with BLR used as a reference. We also asked a native English speaker to read our manuscript; these essential points were perfectly clear to him.
Also, the idea presented as the main result in the discussion and conclusion is about maternal education level associated with incorrect answers about child development. However, only one paragraph of discussion is about this, all the rest of the discussion is about the comparison between the two instruments used.
As detailed in the manuscript, the main result is that, using the BLR as a reference, the ASQ-3 has a low sensitivity and that a low maternal education level is a major risk factor of incorrectly evaluating children with the ASQ-3. Our discussion, containing one paragraph about maternal education, one paragraph about BLR, one paragraph about the performance of the ASQ-3, and one paragraph about strengths and limitations, seems to be well-balanced.
Further, considering low maternal education level is associated with incorrect answers in ASQ-3, the conclusion should state this instrument could not be adequate for THIS population, and not a general population. What are the results for subjects with a high education level?
Thank you for this comment. We enhanced our conclusion to explain that the ASQ-3 should not be used alone to follow the development of preterm children, because (i) the ASQ-3 has a low sensitivity and (ii) a low maternal education level is a major risk factor of incorrectly evaluating children with the ASQ-3.